# Isomer design unlocks rainbow phosphorescence

Xinyue Xu[1], Dong Ding[1], Xinyu Ding[1], Shaoyang Han[1], Erkin Zakhidov[2], Feng Li[3] ✉ & Mingliang Sun [1] ✉

Achieving predictable color-tunable organic room-temperature phosphorescence (RTP) remains challenging due to limited understanding of triplet-state regulation in heteroaromatic systems. Carbazole and benzindole isomers provide an ideal platform to clarify how nitrogen positional isomerism governs triplet exciton behavior and emission energetics. Here, we establish a unified comparative framework to systematically investigate carbazole together with Bd[f], Bd[e], and Bd[g]. Nitrogen-site modulation within the fused tricyclic skeleton generates distinct red, yellow, green, and blue phosphorescence, while mechanochemical solvent-free synthesis enables scalable preparation of previously inaccessible benzindole isomers. Photophysical measurements combined with DFT/TD-DFT calculations, single-crystal analysis, and interaction region indicator theory reveal that positional isomerism controls exciton localization, triplet stabilization, and nonradiative decay independent of the host matrix. Here, we show that isomer-regulated triplet dynamics enable full-spectrum RTP, ultralong lifetimes up to 4.23 s, TSFRET behavior, and matrix-universal multifunctionality, establishing a general molecular design principle for rainbow-like organic phosphorescent materials.

Ultralong organic phosphorescent (UOP) materials have garnered significant interest due to their unique long-lived emission characteristics and broad application prospects in optoelectronics, bioimaging, information encryption, and anti-counterfeiting[1–10]. However, achieving persistent intense phosphorescence under ambient conditions remains a formidable challenge. This limitation primarily arises from the intrinsically weak spin-orbit coupling (SOC) in purely organic systems and the rapid nonradiative decay of triplet excitons. The spin-forbidden nature of intersystem crossing (ISC) between singlet ($S_n$) and triplet ($T_n$) excited states further impedes the generation and utilization of triplet states, thereby compromising RTP efficiency[11–18]. To address these issues, extensive efforts have been devoted to developing molecular design principles and materials engineering strategies aimed at enhancing room-temperature phosphorescence (RTP). Representative approaches include crystal engineering, polymer matrix confinement, host-guest doping systems, molecular self-assembly, and the construction of supramolecular architectures[19–23]. These strategies generally operate by rigidifying the molecular environment, suppressing intramolecular vibrational and rotational motions, minimizing nonradiative decay pathways, and stabilizing the triplet state to prolong exciton lifetimes[24–29]. However, despite these advancements, the precise and predictable modulation of phosphorescence emission wavelengths within a unified molecular framework remains a significant challenge[30–33]. In this context, structural isomerism represents a powerful yet underexplored approach. By altering the connectivity or spatial arrangement of key atoms within the same molecular backbone, isomeric molecules can exhibit distinct electronic structures, intermolecular packing modes, and excited-state dynamics, thereby enabling systematic tuning of energy levels, exciton diffusion behavior, and emission characteristics.

Among the various molecular frameworks employed in RTP materials, carbazole and its derivatives have attracted sustained

[1]School of Materials Science and Engineering, Ocean University of China, Qingdao, China. [2]Institute of Ion-Plasma and Laser Technologies of the Academy of Sciences of the Republic of Uzbekistan, Tashkent, Uzbekistan. [3]Analytical and Testing Center, Qingdao University of Science & Technology, Qingdao, China. ✉e-mail: lifeng02@qust.edu.cn; mlsun@ouc.edu.cn

attention due to their rigid coplanar structures, high triplet energy levels, and excellent photostability[34–38]. As a structural isomer of carbazole, benzindole (Bd) offers additional molecular tunability through variation of the nitrogen atom position within the fused tricyclic aromatic system, giving rise to several regioisomers (Bd[g], Bd[e], and Bd[f]) with distinct electronic structures. Despite their close structural similarity, subtle positional differences between benzindole and carbazole result in markedly different triplet-state behaviors and RTP properties, while benzindole-based RTP systems have received comparatively little attention. It was not until 2020 that Liu[39] reported that trace Bd[f] impurities in commercial carbazoles could significantly influence RTP behavior, which subsequently stimulated growing interest in Bd[f]-based systems[40–42]. Nevertheless, most existing studies remain focused on the Bd[f] configuration[8,43,44], while other isomers such as Bd[e] and Bd[g] have received far less attention, and systematic structure property comparisons across the carbazole-benzindole isomer family are still lacking. Although previous studies have demonstrated that altering the nitrogen atom position can modulate phosphorescence color[45], these investigations were limited to individual molecules rather than a complete isomeric series. In contrast, the present work systematically investigates the full carbazole-benzindole family (Cz, Bd[f], Bd[e], and Bd[g]), enabling unified structure property correlations and full-spectrum RTP color tuning in polymer matrices. Notably, unlike conventional strategies relying on extensive substitution, heavy-atom effects, or multi-component host-guest systems, full-spectrum RTP is achieved here through minimalist backbone isomerization within a single heteroaromatic framework, offering a structurally economical and scalable design approach.

A major challenge in this field lies in the synthesis of certain isomeric forms-particularly Bd[f]-which typically require multi-step solution-based protocols characterized by low selectivity, prolonged reaction times, and cumbersome purification steps. These factors have significantly impeded direct comparative studies among Bd isomers. Recent findings demonstrate that the rarely explored Bd[g] isomer can be selectively synthesized via a one-step, solvent-free mechanochemical approach under ball-milling conditions, offering a greener and more efficient alternative to traditional methods[46]. Building on recent advances in solvent-free mechanochemical synthesis, Bd[g] and Bd[e] derivatives were efficiently obtained via a one-step grinding strategy, while Bd[f]-based analogs were prepared through optimized multistep solution synthesis. This synthetic capability enables the construction of a complete Cz/Bd isomeric platform, including Cz, Bd[f], Bd[e], and Bd[g] (Fig. 1a).

Here, we show that this unified molecular family establishes systematic correlations between isomeric structure and key RTP parameters, including emission color, triplet-state energetics, and lifetime. When dispersed in polymer matrices such as poly(vinyl alcohol) (PVA), these luminophores collectively span the full visible spectrum from blue to red under ambient conditions. Notably, Bd[g] and Bd[e], as isomeric analogs of carbazole, exhibited RTP performance not only in PVA but also in other matrices such as polyvinylpyrrolidone (PVP), poly(vinyl butyral) (PVB), and poly(methyl methacrylate) (PMMA), underscoring the universal nature of their phosphorescent behavior (Fig. 1b). The resulting UOP materials, when embedded in PVB and PMMA matrices, exhibit excellent chemical stability, long-lasting afterglow, and seawater tolerance, enabling practical applications in underwater imaging, durable anti-counterfeiting, and marine bio-interaction studies. Their visible phosphorescence under UV or sunlight activation offers opportunities for optical tagging and remote sensing in complex ocean environments. (Fig. 1c). The spectral tunability demonstrated by this heterogeneous platform provides a good foundation for the development of next-generation RTP technologies.

## Results
### Synthesis of isomers
The carbazole model compound (Ph-Cz) and the benzindole[f] isomer (Ph-Bd[f]) were synthesized using a conventional solution-phase thermal method, a widely adopted approach that typically involves multi-step reactions, extended purification processes, and substantial solvent consumption. The overall yields for Ph-Cz and Ph-Bd[f] were 70% and 16%, respectively. In contrast (Fig. 2a), the less explored isomers Ph-Bd[g] and Ph-Bd[e] were successfully synthesized under solvent-free conditions using a one-step mechanochemical ball-milling strategy, achieving high efficiency and yields of approximately 76% and 46%. This solid-state approach not only simplifies the synthetic

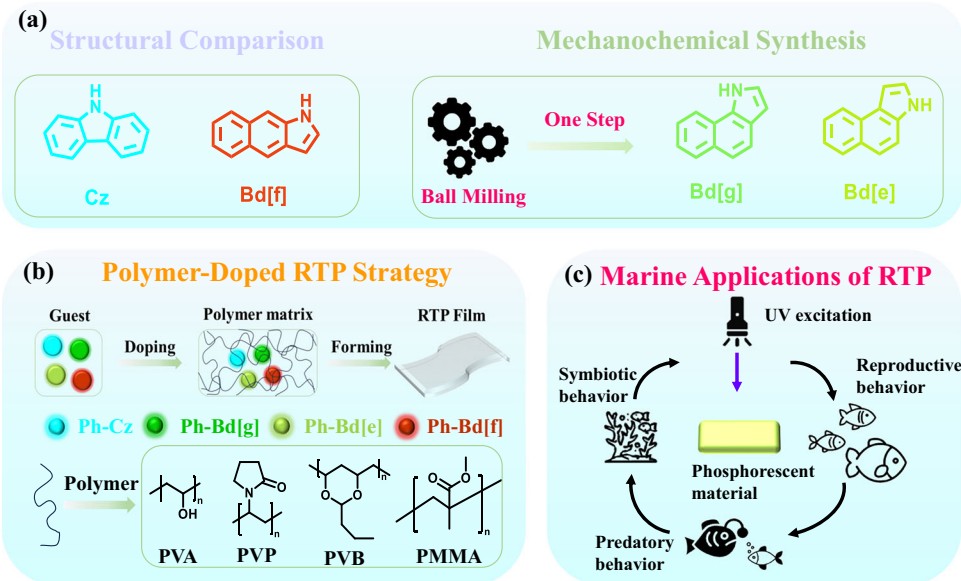

**Fig. 1 | Synthesis, polymer doping, and marine applications. a** Comparison between thermal solvent synthesis of Cz and Bd[f] backbone molecules and mechanochemical ball-milling synthesis of Bd[g] and Bd[e] backbone molecules. **b** Schematic illustration of doping and film formation of four guest molecules (Blue represents Ph-Cz, green represents Ph-Bd[g], yellow-green represents Ph-Bd[e], and red represents Ph-Bd[f]). in four different polymer matrices (PVA, PVP, PVB, and PMMA). **c** Schematic illustration of the potential applications of phosphorescent materials in marine environments.

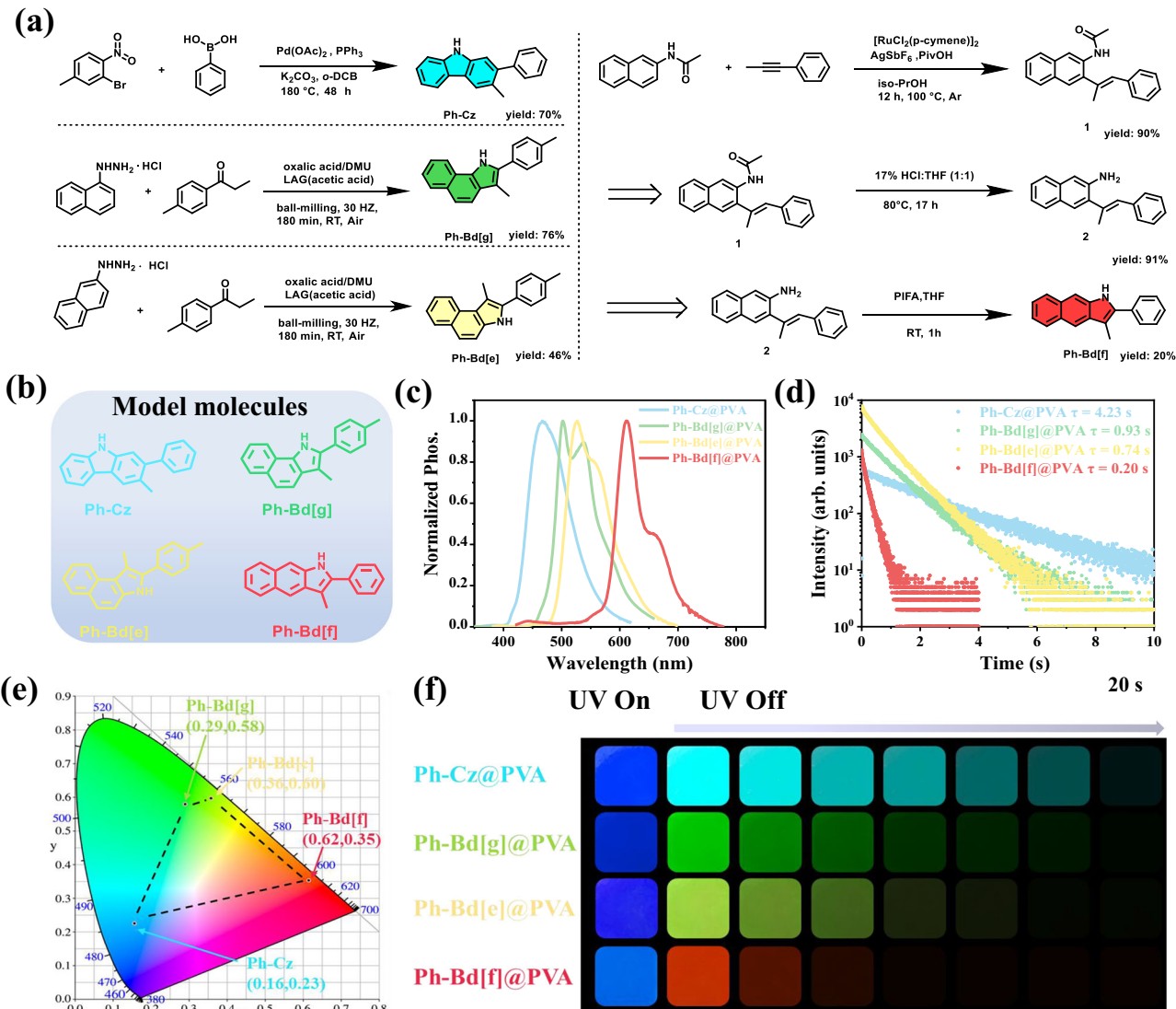

**Fig. 2 | Molecular synthesis and photophysical properties. a** Synthetic routes to four guest molecules (Ph-Cz, Ph-Bd[g], Ph-Bd[e] and Ph-Bd[f]). **b** Chemical structures of the four model carbazole and benzindole molecules. **c** Phosphorescence emission spectra of PVA films doped with each guest molecule (The blue line represents Ph-Cz@PVA, the green line represents Ph-Bd[g]@PVA, the yellow line represents Ph-Bd[e]@PVA, and the red line represents Ph-Bd[f]@PVA). **d** Time-resolved phosphorescence decay profiles of the doped PVA films (The blue line

represents Ph-Cz@PVA, the green line represents Ph-Bd[g]@PVA, the yellow line represents Ph-Bd[e]@PVA, and the red line represents Ph-Bd[f]@PVA). **e** CIE chromaticity diagram showing the tunable afterglow colors of the four RTP systems (Blue represents Ph-Cz, green represents Ph-Bd[g], yellow represents Ph-Bd[e], and red represents Ph-Bd[f]). **f** Photographic images of Ph-Cz@PVA, Ph-Bd[g]@PVA, Ph-Bd[e]@PVA, and Ph-Bd[f]@PVA films under 365 nm UV irradiation (UV On, fluorescence) and after cessation of excitation (UV Off, phosphorescence).

process but also adheres to the principles of green chemistry by eliminating hazardous solvents and minimizing energy input. Bd[g] and Bd[e], representing previously underutilized isomeric cores within the benzindole family, can be accessed via a sustainable and scalable route and exhibit favorable optical properties compared to the extensively studied Bd[f] isomer. These features render Bd[g] and Bd[e] promising candidates for the development of next-generation, heavy metal-free RTP materials. The incorporation of Bd[g] and Bd[e] into the existing carbazole-Bd[f] framework enables a complete and systematic comparison across the isomeric series. The chemical structures and purities of the synthesized compounds were confirmed by nuclear magnetic resonance (NMR) spectroscopy, single-crystal X-ray diffraction (SCXRD), high-resolution mass spectrometry (HR-MS), and high-performance liquid chromatography (HPLC). Detailed synthetic procedures and characterization data are provided in the Supporting Information (Supplementary Figs. 1–4; Supplementary Figs. 5–22). The resulting carbazole and benzindole-based isomers

were physically doped into various polymer matrices at a concentration of 1 wt%, and homogeneous thin films were obtained via drop-casting, as detailed in the Materials Preparation section of the Methods.

## Color-tunable RTP from isomeric guests

The molecular structures of the four guest molecules are presented in Fig. 2b. Differential scanning calorimetry (DSC) measurements indicated that the doped PVA films exhibited similar glass transition and melting temperatures, suggesting that the low doping concentration had negligible influence on the interactions between PVA chain segments, thereby preserving the intrinsic semi-crystalline nature of the polymer matrix (Supplementary Fig. 23b). Thermogravimetric analysis (TGA) further confirmed the excellent thermal stability of all doped films, with decomposition temperatures exceeding 240 °C, highlighting their suitability for practical applications (Supplementary Fig. 23a). Upon cessation of UV excitation, the doped films Ph-Bd[f]

@PVA, Ph-Bd[e]@PVA, Ph-Bd[g]@PVA, and Ph-Cz@PVA exhibited rainbow-like persistent room-temperature phosphorescence (RTP) (Supplementary Fig. 28), featuring distinct red, yellow, green, and blue emissions across the visible spectrum. Their phosphorescence emission maxima were observed at 611 nm, 527 nm, 502 nm, and 467 nm, respectively (Fig. 2c), closely matching the low-temperature phosphorescence profiles of the corresponding guest molecules measured at 77 K (Supplementary Fig. 25). No room-temperature phosphorescence was observed for the guest molecules in their pristine states. (Supplementary Fig. 26) The CIE chromaticity coordinates of the RTP emissions were determined to be (0.62, 0.35), (0.36, 0.60), (0.29, 0.58), and (0.16, 0.23), respectively (Fig. 2e). As illustrated in Fig. 2f, photographic images captured after switching off the UV irradiation revealed distinct blue, green, yellow, and red emissions from Ph-Cz@PVA, Ph-Bd[g]@PVA, Ph-Bd[e]@PVA, and Ph-Bd[f]@PVA, respectively-corresponding to the four primary colors essential for display and full-color applications. To date, this represents the demonstration of full-spectrum RYGB phosphorescence emission from purely organic phosphors through a simple molecular backbone isomerization strategy.

In order to gain insight into the phosphorescence behavior of these RTP films containing carbazole- and benzindolyl-guest molecules with different configurations, their photophysical parameters, including phosphorescence lifetimes and quantum yields, were systematically investigated as shown in Fig. 2d and Supplementary Fig. 27. Notably, the Ph-Cz@PVA film exhibits RTP performance, featuring both a high phosphorescence quantum yield ($\Phi_P = 16.6\%$) and an ultralong lifetime ($\tau_P = 4.23$ s). In comparison, although the Ph-Bd-based systems do not reach the same level of RTP efficiency and lifetime as Ph-Cz, they offer unique advantages as structural isomers. These isomers enable tunable phosphorescence emissions across a wide spectral range, from green, yellow to red, via precise control over molecular configuration. Specifically, Ph-Bd[g]@PVA and Ph-Bd[e]@PVA demonstrate bright green and yellow RTP emissions, respectively ($\tau_P = 0.93$ s, $\Phi_P = 13.4\%$; $\tau_P = 0.74$ s, $\Phi_P = 12.9\%$). Despite their relatively shorter lifetimes, these isomers are synthetically more accessible and cost-effective, making them attractive candidates for scalable applications in areas such as anti-counterfeiting and optical tagging. On the other hand, Ph-Bd[f]@PVA displays a red afterglow emission with similar quantum efficiency ($\Phi_P = 10.7\%$) but a modest lifetime ($\tau_P = 0.2$ s) which agree well with other Bd[f] phosphor[8,39,47]. However, the synthesis of the Bd[f] isomer remains more challenging and less economically viable, limiting its practical applicability in large-scale material development. The contrasting performance profiles of Ph-Cz and its benzindole isomers provide a valuable framework for understanding the structure-property relationships governing organic phosphorescence. Further mechanistic studies are warranted to elucidate the underlying photophysical pathways, particularly the factors contributing to the long-lived RTP of Ph-Cz and the color tunability observed across the Bd-series.

**Mechanism of isomer RTP**

To gain deeper insight into the room-temperature phosphorescence (RTP) emission mechanisms of carbazole and its benzindole derivatives, systematic density functional theory (DFT) and time-dependent DFT (TD-DFT) calculations were performed at the B3LYP/6-31 G level (Supplementary Data. 1). As shown in Fig. 3a, the calculated HOMO-LUMO energy gaps exhibit a progressively decreasing trend with structural isomerism, namely Ph-Cz (4.56 eV), Ph-Bd[e] (4.34 eV), Ph-Bd[g] (4.11 eV), and Ph-Bd[f] (3.65 eV), which correlates well with the experimentally observed red-shifts in the absorption (Supplementary Fig. 29) and phosphorescence spectra (Fig. 2b). This trend can be attributed to the enhanced π-conjugation introduced by the benzindole moieties[48], which effectively reduces the electronic band gap and lowers the triple energy level ($T_1$) (Supplementary Fig. 30a), as

evidenced by the cavity/electron composition analysis, where Ph-Cz exhibits the smallest charge density difference during the $T_1$ transition (Supplementary Figs. 39–42). To further evaluate the efficiency of the intersystem crossing (ISC) process, the singlet and triplet energy level distributions and spin-orbit coupling (SOC) constants within a ± 0.3 eV energy window around the $S_1$ state were analyzed. Considering the realistic operational environment of the materials, a polyvinyl alcohol (PVA) segment was introduced as a model system to optimize the conformations of the guest molecules embedded within the polymer matrix (Fig. 3b). All four compounds exhibit small $S_1$-$T_n$ energy gaps and relatively high SOC constants, favoring efficient ISC transitions (Supplementary Tables. 2–5). Notably, the calculated results in both gas-phase and crystal-state structures are highly consistent (Supplementary Fig. 30), further confirming the generality and structure-governed nature of the observed electronic evolution. Furthermore, Hole-Electron analyses were performed between the $S_1$ state and the representative $T_n$ states with the highest transition contributions (Supplementary Figs. 43–44) to quantitatively elucidate the excited-state charge-transfer characteristics. The $S_1$ states of all four systems are predominantly governed by HOMO-LUMO π-π* transitions, with contributions of 91.0% for Ph-Cz, 71.4% for Ph-Bd[e], 96.0% for Ph-Bd[g], and 97.3% for Ph-Bd[f]. According to El-Sayed's rule[49], the presence of triplet states with significant n-π* character can facilitate spin-flip processes. Correspondingly, the $T_5$ state of Ph-Cz (54.4%), $T_6$ of Ph-Bd[g] (95.5%), $T_5$ of Ph-Bd[e] (95.8%), and $T_2$ of Ph-Bd[f] (63.4%) all exhibit prominent n-π* characteristics, theoretically supporting efficient $S_1$-$T_n$ ISC channels. To account for dynamic host-guest effects, full molecular dynamics (MD) simulations were performed for all mixed systems, followed by electronic structure analysis of the MD-sampled conformational ensembles. Across all systems, the $T_1$ energy levels extracted from the MD trajectories consistently follow a decreasing order from Cz to Bd[e], Bd[g], and Bd[f], in agreement with the static DFT results, indicating that the triplet-state energetic evolution is primarily governed by backbone isomerism (Supplementary Figs. 31–38).

Despite theoretically favorable spin-orbit coupling (SOC) constants and efficient intersystem crossing (ISC) channels predicted for the isomeric systems, the experimentally observed room-temperature phosphorescence (RTP) lifetimes of the benzindole derivatives are significantly shorter than that of Ph-Cz. In practical systems, molecular rigidity, matrix-imposed restriction of intramolecular motions, and synergistic non-covalent interactions-particularly hydrogen bonding-are essential for suppressing nonradiative decay and stabilizing triplet excitons. Detailed electrostatic potential (ESP) surface analyses[50] were performed based on density functional theory (DFT) and Multiwfn[51–53] calculations to investigate the interactions between the phosphorescent guest molecules and the PVA matrix (Fig. 3c). All four luminophores exhibit distinct ESP distributions, with negative potential localized on their aromatic cores and positive potential near the N-H groups. Notably, while the negative ESP values for all guests are comparable (~ −20 kcal/mol), their positive ESP values vary significantly, with Ph-Cz exhibiting the highest value (+45.15 kcal/mol). This gradient corresponds to a progressive reduction in intramolecular charge separation across the series from Ph-Cz, Ph-Bd[e], Ph-Bd[g] to Ph-Bd[f]. These spatial charge distributions promote electrostatic complementarity with the hydroxyl-rich PVA matrix, enhancing host-guest interactions that facilitate microenvironmental rigidification. The pronounced electrostatic polarity of Ph-Cz leads to stronger binding with the polymer host, effectively restricting molecular motions and suppressing nonradiative deactivation pathways. This theoretical insight aligns well with the experimental RTP lifetime data (Fig. 2c), where Ph-Cz@PVA displays the longest phosphorescence lifetime among the series, underscoring the pivotal role of electrostatic compatibility in stabilizing long-lived triplet states. Collectively, these multiscale

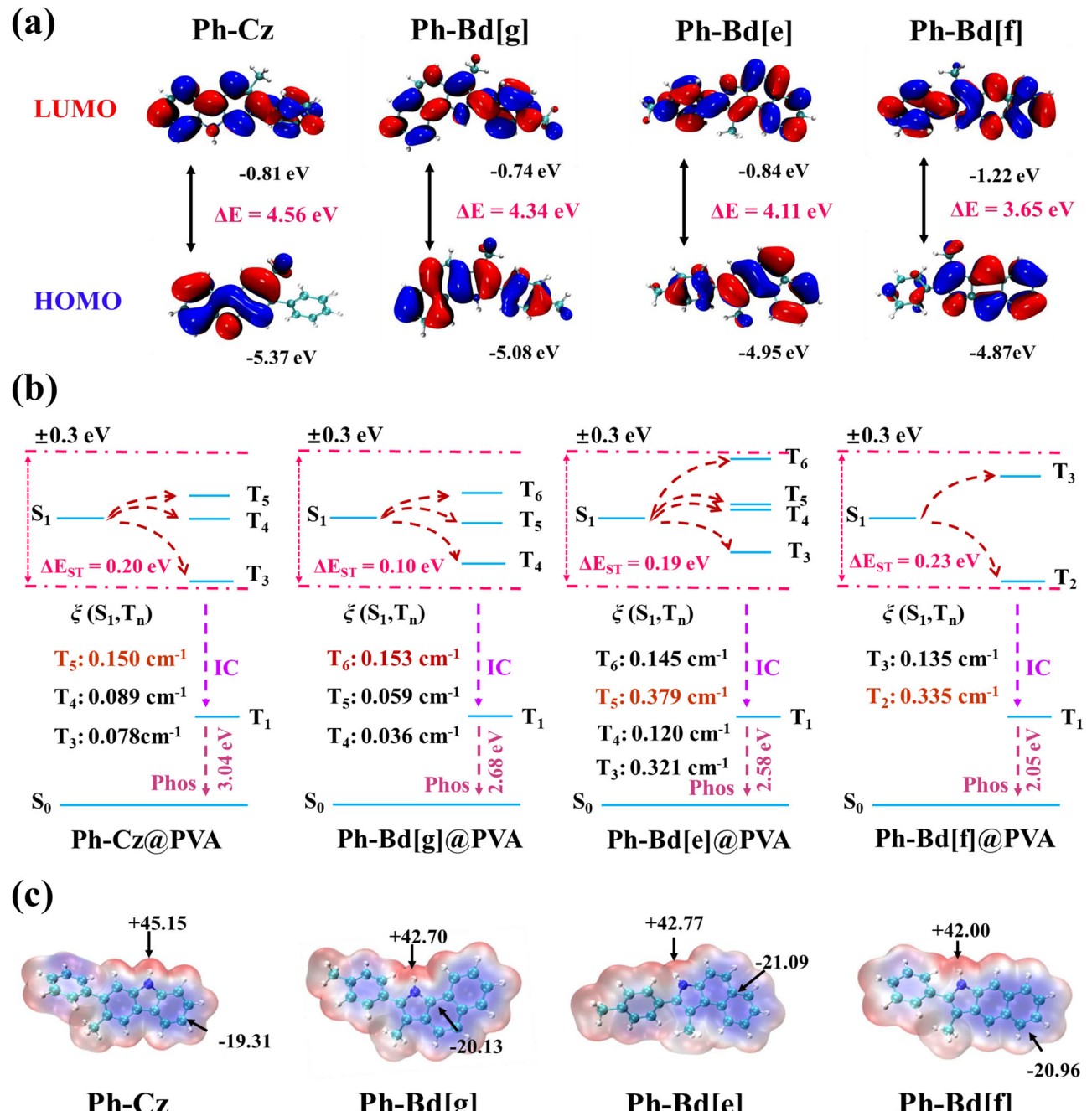

**Fig. 3 | Electronic structure and ISC of carbazole and benzindole RTP. a** HOMO and LUMO orbital distributions and calculated energy gaps of the four guest molecules. **b** Schematic energy level diagrams and spin-orbit coupling (SOC) constants of the corresponding PVA-doped systems, with potential intersystem crossing (ISC) pathways indicated by red arrows. **c** Electrostatic potential (ESP) surface maps of the four guest molecules (Ph-Cz, Ph-Bd[g], Ph-Bd[e], and Ph-Bd[f]), illustrating charge distribution characteristics relevant to host-guest interactions.

investigations highlight that efficient RTP arises from a synergistic interplay among electronic structure.

**Mechanistic understanding of carbazole-based RTP**

To further understand the origin of the RTP performance of Ph-Cz, we conducted structural-level analyses focusing on its intrinsic molecular geometry and solid-state packing behavior. Single crystals of guest were obtained via slow evaporation from a dichloromethane/n-hexane solvent mixture under ambient conditions (Supplementary Figs. 20–22). Subsequent single-crystal X-ray diffraction analysis revealed that Ph-Cz crystallizes in the orthorhombic $P2_12_12_1$ space group (Supplementary Table 1), adopting a distinct parallel-stacked dislocation motif. The crystal lattice features a dense network of noncovalent interactions, including N-H...C hydrogen bonds (2.80–2.88 Å) and pronounced π...π stacking ( ~ 2.93 Å) between adjacent carbazole units (Fig. 4a), which collectively promote exciton confinement. The asymmetric unit contains four independent molecules, providing multiple rigidified conformations (Supplementary Fig. 20). A large dihedral angle (51.6°) between the carbazole core and the pendant phenyl ring induces a twisted, non-planar geometry, which suppresses nonradiative decay while maintaining favorable spin-orbit coupling in the solid state. To further elucidate the mechanism underlying the RTP

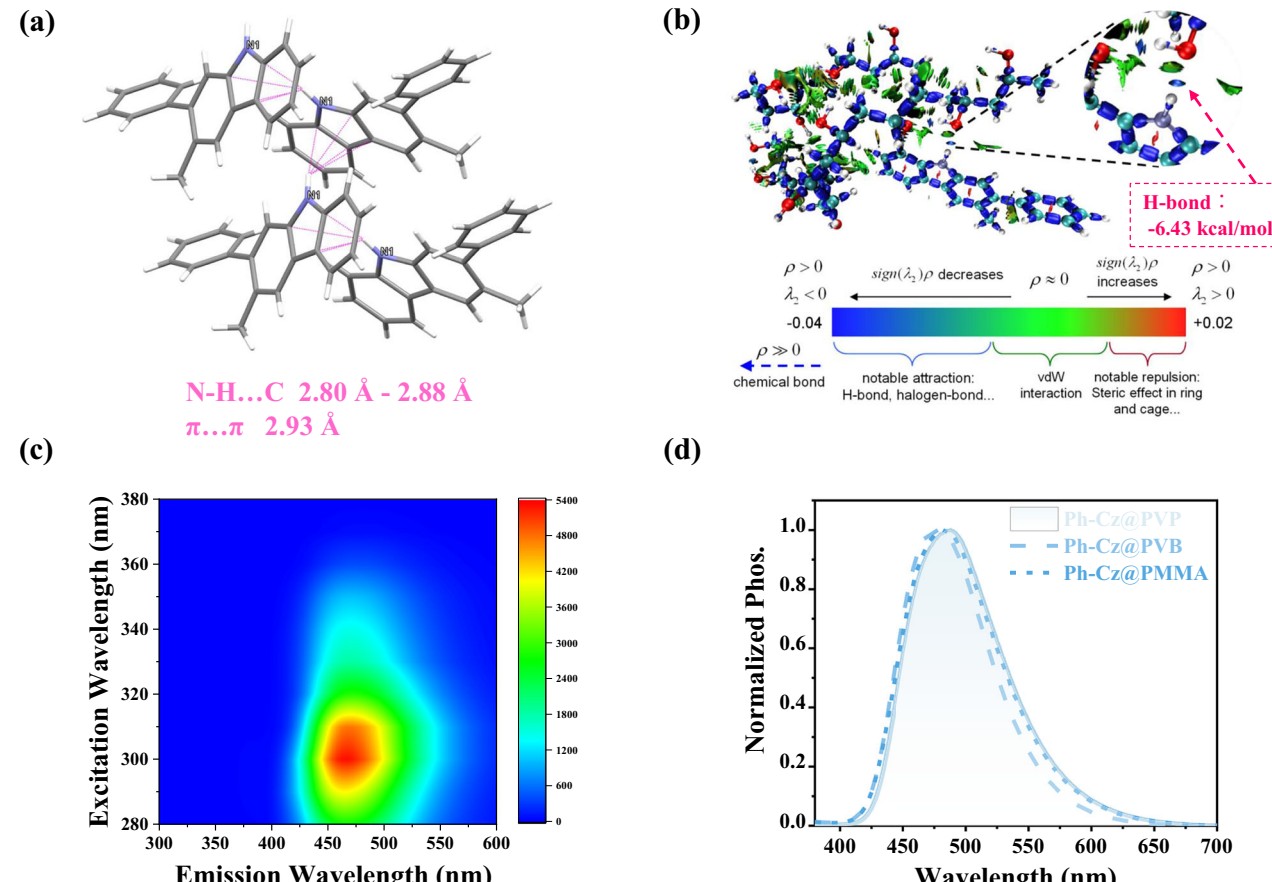

N-H···C  2.80 Å - 2.88 Å
π···π   2.93 Å

H-bond :
-6.43 kcal/mol

**Fig. 4 | Crystal structure, interactions, and phosphorescence of Ph-Cz in polymers. a** Single-crystal structure of Ph-Cz, showing the parallel-stacked dislocation motif and key intermolecular interactions, including N-H···C hydrogen bonds and π···π stacking. **b** Interaction region indicator (IRI) analysis revealing non-covalent interactions between Ph-Cz and the PVA matrix, including hydrogen bonding and van der Waals forces. **c** Excitation-phosphorescence contour map of the Ph-Cz@PVA film. **d** Phosphorescence emission spectra of Ph-Cz doped into various polymer matrices (PVP, PVB, and PMMA).

performance of Ph-Cz in polymer matrices[54], interaction region indicator (IRI) analysis was performed using Multiwfn[52,53,55] to visualize the non-covalent interactions between Ph-Cz and the PVA host (Fig. 4b). In addition to extensive van der Waals contacts, Ph-Cz forms strong directional hydrogen bonds between its N-H group and the hydroxyl groups of PVA, with a calculated interaction energy of −6.43 kcal mol⁻¹. These interactions effectively rigidify the local microenvironment, suppress nonradiative deactivation, and stabilize triplet excitons. In contrast, the benzindole derivatives Ph-Bd[g] and Ph-Bd[e] interact with PVA predominantly through weak van der Waals forces, while Ph-Bd[f] exhibits only limited hydrogen-bonding capability (Supplementary Fig. 45), rationalizing their comparatively lower RTP efficiencies.

Excitation-phosphorescence mapping of Ph-Cz@PVA shows excitation-independent emission over a broad range (Fig. 4c), indicating homogeneous dispersion and a single emissive species. To assess matrix dependence, Ph-Cz was further embedded in PVP, PVB, and PMMA, polymers with distinct rigidity, polarity, and oxygen permeability. Persistent RTP is observed in all matrices (Fig. 4d, Supplementary Fig. 46, Supplementary Table 6), confirming the intrinsic photophysical robustness of Ph-Cz. However, the exceptionally prolonged lifetime is unique to PVA, highlighting the critical role of hydrogen-bond-assisted confinement in triplet stabilization.

### Multicolor afterglow via TS-FRET
Guided by the mechanistic benchmark established by Ph-Cz, and considering both the synthetic challenges associated with the Bd[f] isomer and the absence of markedly improved phosphorescent

performance in the Ph-Bd[f]@PVA system, attention was directed toward the more synthetically accessible and functionally advantageous Ph-Bd[g] and Ph-Bd[e] derivatives. In benzindole isomers, nitrogen-position isomerization intrinsically reshapes triplet-state energies, providing an effective handle for tuning RTP emission wavelengths without relying on substituents or matrix effects. Ph-Bd[g] and Ph-Bd[e] were doped into various polymer matrices, including PVP, PVB, and PMMA, to achieve guest-induced organic room-temperature phosphorescence (Fig. 5a), with their corresponding quantum yields and lifetimes summarized in Supplementary Figs. 47–48. The resulting doped systems exhibit distinct and persistent RTP emissions (Supplementary Table 6), establishing Ph-Bd[g] and Ph-Bd[e] as robust phosphorescent scaffolds for wavelength-tunable organic afterglow.

These systems show robust room-temperature phosphorescence, indicating their promising potential for tunable afterglow emission via triplet-to-singlet Förster resonance energy transfer (TSFRET). To this end, two commercially available fluorescent dyes-fluorescein sodium (FLS) and rhodamine B (RHB)-were selected as energy acceptors (Fig. 5f). Their absorption spectra show substantial overlap with the phosphorescence emission spectra of Ph-Bd[g]@PVA and Ph-Bd[e]@PVA (Fig. 5b), fulfilling the spectral matching requirements for efficient TSFRET. Upon doping FLS and RHB into these RTP matrices, ternary composite films were prepared. Delayed emission spectra of the resulting materials display distinct emission peaks at 555 nm and 620 nm (Fig. 5c), confirming successful energy transfer. Further evidence is provided by CIE chromaticity analysis (Fig. 5e), which

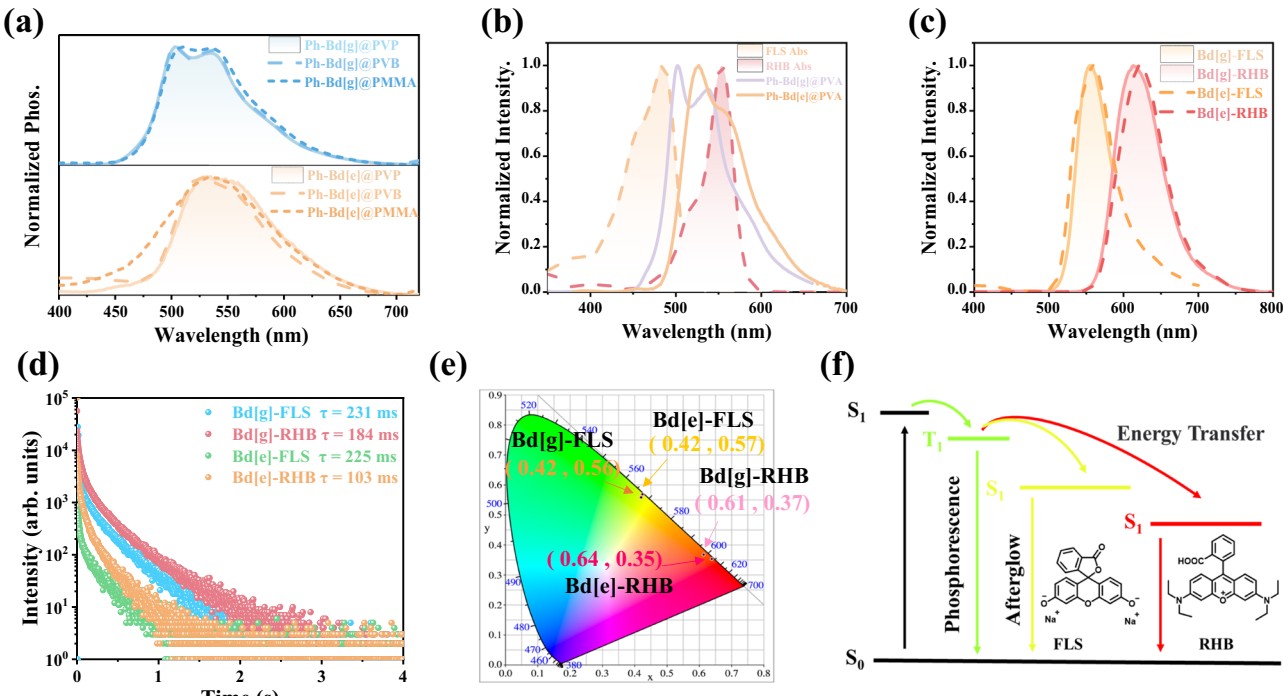

**Fig. 5 | Phosphorescence and FRET in Bd[g] and Bd[e] doped polymer systems. a** Phosphorescence emission spectra (Blue represents Ph-Bd[g], orange represents Ph-Bd[e]) of Ph-Bd[g] and Ph-Bd[e] doped into various polymer matrices (PVP, PVB, and PMMA). **b** Normalized absorption spectra of FLS- and RHB-doped films, along with the phosphorescence spectra of Ph-Bd[g]@PVA and Ph-Bd[e]@PVA (The orange dashed line and red dashed line separately represent FLS and RHB. The purple solid line and orange solid line respectively represent Ph-Bd[g]@PVA and Ph-Bd[e]@PVA). **c** Delayed emission spectra of Bd[g]-FLS, Bd[g]-RHB, Bd[e]-FLS,

and Bd[e]-RHB systems (Orange solid line, red solid line, orange dashed line, red dashed line respectively represent Bd[g]-FLS, Bd[g]-RHB, Bd[e]-FLS, Bd[e]-RHB). **d** Time-resolved luminescence decay curves of the corresponding donor-acceptor systems (Blue, red, green, and orange represent Bd[g]-FLS, Bd[g]-RHB, Bd[e]-FLS, and Bd[e]-RHB, respectively). **e** CIE 1931 chromaticity diagram illustrating afterglow color tuning. **f** Schematic illustration of the FRET process between RTP donors and fluorescent acceptors.

indicates a significant change in the afterglow color after energy transfer. Even after the 365-nm ultraviolet excitation was stopped, persistent yellow and red luminescence could still be observed with the naked eye (Supplementary Fig. 49). Time-resolved decay curves (Fig. 5d) show that the afterglow lifetimes of Bd[g]-FLS, Bd[g]-RHB, Bd[e]-FLS and Bd[e]-RHB, are 281 ms, 184 ms, 225 ms, and 103 ms, respectively, highlighting the effective energy collection and transfer processes of these materials. Overall, these results underscore the advantages of the Bd[g] and Bd[e] isomers-both overlooked structural analogs of carbazole-over the Bd[f] isomer. While the FRET process has been reported for RTP systems[56], here the Bd[g] and Bd[e] isomers enable more synthetically accessible and tunable multicolor afterglow, providing opportunities for designing ternary energy-transfer systems with controlled emission. Their benefits extend beyond synthetic accessibility to compatibility with time-resolved Förster resonance energy transfer (TSFRET)-based strategies, enabling more efficient and tunable multicolor organic afterglow emissions.

## Functional applications

All four guest molecules demonstrated excellent phosphorescent emission across various polymer matrices. By leveraging the distinct physicochemical properties of each host polymer, their application potential in functional RTP systems was systematically investigated. Notably, when doped into a polyvinylpyrrolidone (PVP) matrix, the guest molecules retained stable room-temperature phosphorescence even under elevated temperatures (up to 100 °C) (Supplementary Fig. 50). Utilizing this property, a high-temperature phosphorescence-based anti-counterfeiting display was constructed by arranging the digits "2025" using different RTP composites (Fig. 6a), including Ph-Cz@PVP for the digit "2", Ph-Bd[g]@PVP for "0", and Ph-Bd[e]@PVP

for "5". Under 365 nm UV irradiation at 100 °C, the assembled pattern displayed uniform blue fluorescence. After removal of the excitation source, the digits exhibited distinct persistent phosphorescence-blue for "2", green for "0", and yellow for "5". This proof-of-concept demonstration highlights the potential of thermally stable organic phosphorescent materials in advanced anti-counterfeiting technologies requiring high-temperature tolerance and multicolor emission encoding. In addition, the Ph-Bd[g]@PVA composite exhibited sunlight-activated RTP behavior (Fig. 6b), offering practical advantages for low-power emergency signage. Following exposure to solar irradiation, the material was transferred into a dark enclosure, where a bright and long-lived green afterglow was observed. The emitted light was sufficiently intense to clearly illuminate an emergency exit sign, ensuring visibility under low ambient lighting conditions. These results underscore the applicability of sunlight-responsive organic RTP materials in energy-independent and self-sustaining safety signage systems. Further demonstrating the practical versatility of these RTP materials, phosphorescent emblems of Ocean University of China and its School of Materials Science and Engineering were fabricated using Ph-Cz@PVA and Ph-Bd[e]@PVA, respectively (Fig. 6c). Under ambient fluorescent lighting, the films remained highly transparent, causing minimal optical interference. Upon cessation of UV excitation, the emblems emitted distinct afterglow-blue for Ph-Cz@PVA and yellow for Ph-Bd[e]@PVA-clearly delineating the institutional logos. This demonstration highlights not only the esthetic adaptability of the materials but also their potential in multifunctional display technologies, signage applications, and low-energy identity lighting systems.

After immersion in natural seawater for one month, the phosphorescent guest molecule-doped PVB and PMMA composites exhibited excellent structural and optical stability. Although some salt

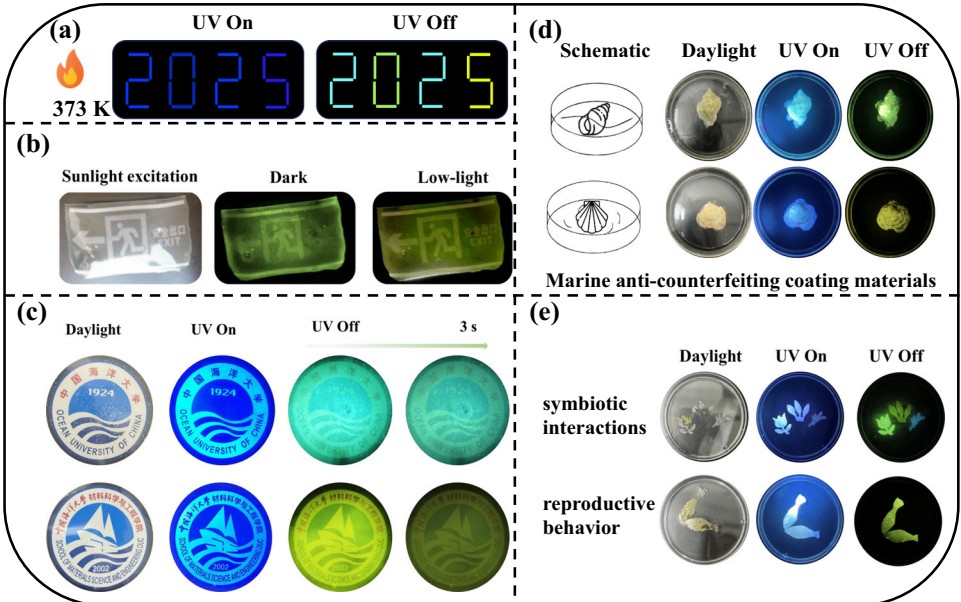

**Fig. 6 | Phosphorescent materials for anti-counterfeiting and marine applications. a** High-temperature anti-counterfeiting pattern "2025" fabricated using phosphorescent materials based on guest molecules doped in a PVA matrix. **b** Phosphorescent illumination of Ph-Bd[g]@PVA material activated by sunlight. **c** Phosphorescent illumination of Ph-Cz@PVA and Ph-Bd[e]@PVA materials activated by UV lamp excitation. **d** Marine anti-counterfeiting coatings fabricated from Ph-Bd[g]@PVB and Ph-Bd[e]@PVB phosphorescent materials. **e** Phosphorescent materials prepared by doping guest molecules into a PMMA matrix to mimic marine biological symbiosis and courtship behaviors.

crystallization was observed due to water evaporation (Supplementary Figs. 51–53), no significant degradation in phosphorescence performance was detected, indicating chemical robustness and corrosion resistance (Supplementary Figs. 54–55). These properties suggest strong suitability for long-term deployment in marine environments. As illustrated in Fig. 6d, coatings of Ph-Bd[g]@PVB and Ph-Bd[e]@PVB applied to shell surfaces maintained clearly visible phosphorescence under UV excitation, even in seawater. This observation confirms the feasibility of utilizing such RTP materials for visually identifiable and persistent luminescent tagging systems on marine equipment, vessels, and offshore platforms. In addition to the demonstrated material stability, the potential relevance of the emission characteristics of these RTP materials to marine biological systems was considered from a photophysical perspective. Many marine organisms exhibit wavelength dependent visual sensitivity; accordingly, the photoluminescence profiles of PMMA-based RTP films were examined under UV excitation (Fig. 6e). The materials display stable and persistent afterglow across the visible region, encompassing wavelength ranges commonly involved in marine visual perception. Notably, the green-to-yellow emission band (approximately 500–580 nm) overlaps with reported sensitivity windows of certain reef-associated fish and benthic invertebrates. This spectral overlap suggests that such RTP materials could provide a useful optical basis for future investigations into light-organism interactions in marine environments.

## Discussion

In summary, we establish a unified carbazole-benzindole isomeric platform (Cz, Bd[f], Bd[e], and Bd[g]) that enables a systematic investigation of how backbone-level isomerism governs organic room-temperature phosphorescence. Precise variation of the nitrogen position within a conserved fused tricyclic framework leads to predictable changes in molecular packing, triplet-state electronic structure, and exciton localization, resulting in continuous RTP color tuning across the full visible spectrum. The emergence of red, yellow, green, and blue (RYGB) phosphorescence arises directly from isomer-dependent triplet-state reorganization, revealing an intrinsic structure-photophysics relationship within the carbazole-benzindole family. A green, one-step high-yeild mechanochemical synthesis was developed for the efficient preparation of Bd[g] and Bd[e] isomers, significantly expanding the structural diversity of carbazole and its benzindole isomer organic phosphorescent molecules. Comparative studies in representative polymer matrices show that the host mainly suppresses nonradiative decay and provides mechanical confinement, whereas the spectral ordering and energetics of RTP emission are determined at the molecular level. Crystallographic, photophysical, and theoretical results consistently indicate that isomerism dominates triplet exciton behavior, with matrix effects primarily influencing emission intensity and lifetime. Beyond mechanistic understanding, the resulting composites exhibit stable and long-lived phosphorescence, including resistance to prolonged seawater immersion, supporting their applicability in information security, optical signaling, and low-energy visual display systems. This work underscores the crucial role of molecular isomerism and polymer matrix synergy in tuning RTP performance and provides an effective strategy for the rational design and functional development of full-color organic phosphorescent materials.

## Methods

### Materials

1,4-Dibromo-2-methyl-5-nitrobenzene (98%), Palladium (II) Acetate (98%) and N-(Naphthalen-2-yl)acetamide (99.9%) were purchased from Adamas Co., Ltd. Silver hexafluoroantimonate (99.9%), 1-Phenyl-1-propyne (99.9%), Dichloro(p-cymene)ruthenium(II) dimer (98%), Oxalic acid (99.58%), 1,3-Dimethylurea (99.96%), Pivalic acid (99.72%) and 1-Naphthylhydrazine hydrochloride (98%) were purchased from Bide Pharmatech Co., Ltd. 4'-Methylpropiophenone (97%) and 2-Naphthylhydrazine hydrochloride (98%) were purchased from Energy Chemical. [Bis(trifluoroacetoxy)iodo]benzene (98%) was purchased from Leyan.com. Triphenylphosphine was purchased from Franklin C. McLean.

## Physical measurements and instrumentation

$^1$H NMR and $^{13}$C NMR were recorded on a Bruker AVANCE III 600 NMR instrument with solvents CDCl$_3$ and DMSO-$d_6$, internal standard tetramethylsilane (TMS), and room temperature. ESI high-resolution mass spectrometry analyses were performed on a Waters Xevo G2 Qt mass spectrometer. Fluorescence and phosphorescence photoluminescence spectra were measured on a Hitachi F-4700 fluorescence spectrometer. Lifetime curves were measured on an FLS1000 photoluminescence spectrometer in Edinburgh. Absolute PL quantum yield (PLQY) was measured on a C11347 integrating sphere spectrometer (Hamamatsu, Japan). UV-visible absorption spectra were measured by a UV-3600 Shimadzu spectrophotometer. Thermogravimetric (TGA) and Differential Scanning Calorimetry (DSC) analyses on the Thermal Analyser STA 8000. High performance liquid chromatography (HPLC) tests were performed on a chromatograph with instrument model SDANA-PC, using a column Shim-pack GIST C18, column temperature of 40 °C, injection volume of 10 μl and flow rate of 1 ml/min. The time interval for data acquisition was set to 500 ms, and the analysis start and end times were from 0 to 20 min, the intensity unit was millivolt (mV), and the intensity multiplier was 0.001. The mechanical ball milling equipment model is XQMCM-2.

## Single crystal analysis

Single Crystal X-ray diffraction data were collected using a Bruker D8 Quest diffractometer (Cu Kα, λ = 1.54178 Å). Indexing and data integration were performed using APEX4 (Difference Vectors method). Absorption correction was performed by multiscan method implemented in SADABS. Space groups were determined using XPREP implemented in APEX4. Structures were solved using SHELXL-2014 (direct methods) and refined using SHELXL-2014 (full-matrix least-squares on F$^2$) with anisotropic displacement contained in APEX4 program packages. Hydrogen atoms on carbon and nitrogen were calculated in ideal positions with isotropic placement parameters set to 1.2 × $U_{eq}$ of the attached atoms.

## Density Functional Theory (DFT) calculations and time-dependent DFT calculations

All chemical structures were optimized at the level of B3LYP/6-31 G and the excited energies were calculated by the time-dependent density functional theory (TD-DFT) method at the level of B3LYP at the 6-31 G (d, p). All the calculations were performed within Gaussian 09 software package. Hole/Particle analyses, atomic charge distributions and electron density surfaces for electrostatic potential (ESP) residency mapping were calculated using Multiwfn software and visualised using the VMD program.

## Preparation of RTP materials

All solvents were purchased from the market and used as is without further purification. Tetrahydrofuran solution (1 ml) of guest molecule (1 mg) was thoroughly mixed with aqueous PVA solution (100 mg PVA1799 (polymerization 17, alcohol solubility 99) and 9 ml water), respectively, then the well mixed solution was dropped into a dye cartridge and baked for 2–3 h at 60 °C to obtain the films. A dichloromethane solution (1 ml) of the guest molecule (1 mg) was thoroughly mixed with a dichloromethane solution (4 ml) of PVP, PVB, and PMMA (100 mg), respectively, and then the well-mixed solution was dropped into a dye cartridge and evaporated to dryness at room temperature to obtain the films. All films were then dried in a vacuum oven at 65 °C for 12 h to remove residual moisture for subsequent testing.

## Data availability

Source data are provided with this paper. The data are available from the corresponding authors upon request. The X-ray crystallographic coordinates for structures reported in this study have been deposited at the Cambridge Crystallographic Data Center and are available free of charge with the following codes: Ph-Cz (CCDC 2466598), Ph-Bd[g] (CCDC 2466760), Ph-Bd[e] (CCDC 2360172). Source data are provided with this paper.

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

## Acknowledgements

The authors are deeply grateful to the National Natural Science Foundation of China (No. 52173291 to M. S.), the Shandong Provincial Natural Science Foundation (No. ZR2024ME032 and ZR2024ZD15 to M. S.), the National Key Research and Development Program of China (No. 2019YFC0312101 to M. S.), and the Key R&D Program of Shandong Province, China (No. 2023CXGC010315 to M. S.).

## Author contributions

The idea and design of the study was proposed by M.L. and X.Y.X., X.Y.D. and S.Y. assisted X.Y.X. with synthetic molecular experiments and data analysis. X.Y.X. and D.D. performed theoretical calculations. X.Y.X. wrote the manuscript. L.F. provided the test bed and in-depth discussion of the data and the manuscript. M.L. provided theoretical guidance, financial support and in-depth discussion of the data and the manuscript. The manuscript is discussed in depth by Erkin Zakhidov. All authors discussed and commented on the manuscript and contributed to the writing of the manuscript.

## Competing interests

The authors declare no competing interests.
