## [Transparent Peer Review file · Nature Communications]

Isomer Design Unlocks Rainbow Phosphorescence

Corresponding Author: Professor Mingliang Sun

Version 0:

Reviewer comments:

Reviewer #1

(Remarks to the Author)

Room-temperature phosphorescence (RTP) is a key aspect and frontier in photophysics. Herein, the authors report an isomeric design strategy for achieving full-color, rainbow-like organic phosphorescence. Precise nitrogen-site isomerization species are of interest for developing state-of-the-art models to address the thorny challenges in the RTP field. However, the manuscript fails to address the key aspect of deep structure-photophysical relationships. Up to date, larger strategies are proposed to tune the emission color of organic RTP materials, which is not the challenge in this area. The intrinsic exciton behaviors of the isomers require further investigation, rather than focusing solely on doping in matrices such as PVA—these matrices are not new for RTP and rely on basic host-guest systems. Additionally, carbazole (Cz), as a foundational reference, should be included for detailed discussion of RTP emission mechanisms. While the four isomers represent a valuable set, their utility is limited to tuning emission wavelength, with only modest, unsystematic changes in RTP efficiency. Critically, this study does not clarify the significant differences between the three benzindole isomers and Cz. I recommend rejection. Specific detailed concerns are as follows:

1. Lack of deep structure-photophysical relationships, alongside the absence of quantitative analysis and generalizable rules for doping matrices. The quantitative effects of isomerism on RTP performance must be further elaborated in the study.
2. The theoretical framework should incorporate molecular dynamics simulations of mixed samples, followed by calculations of electronic structures to better reflect real-world system dynamics.
3. the key role of non-novel matrices (e.g., PVA) and incomplete analysis of the host-guest system, which fails to advance beyond established RTP matrix design paradigms.

Given these fundamental limitations, I recommend rejecting the manuscript. To strengthen the work, the authors should: (1) dissect the intrinsic exciton behaviors of the isomers (e.g., via comparative studies of neat isomer films vs. polymer-doped systems); (2) integrate molecular dynamics (MD) simulations into the theoretical analysis to model dynamic isomer-matrix interactions; (3) establish quantitative structure-RTP relationships (e.g., linking nitrogen position to triplet energy levels or intersystem crossing efficiency); and (4) frame Cz as a mechanistic benchmark to derive clear rules for distinguishing benzindole isomers. A revised version of this work may be more suitable for submission to specialized journals.

Reviewer #2

(Remarks to the Author)

The paper presents a comprehensive and systematic study of the room-temperature phosphorescence (RTP) properties of carbazole (Cz) and its benzindole isomers (Bd[f], Bd[e] and Bd[g]). By precisely controlling the nitrogen atom position within the tricyclic backbone, the authors achieve full-spectrum, tunable RTP spanning red, yellow, green, and blue emissions. Two new isomers, Bd[e] and Bd[g] were synthesized by a one-pot solvent-free mechanical method, which is greener and more efficient than the traditional multi-step solution method (Bd[f]). The material exhibits excellent stability and long afterglow emission in matrices such as PVA, PVP, PVB, and PMMA, and can be used in applications such as anti-counterfeiting, Marine identification, and biological simulation.

The experimental content is innovative and has a complete experimental system. However, there are numerous issues in writing and formatting. I recommend the manuscript requires a thorough revision before it can be considered for publication.

1. The paper reads overly complicated and repetitive in many parts, suggesting that the authors may have relied heavily on AI-assisted writing. I recommend a substantial revision to improve clarity and authenticity of expression.
2. The NMR spectra don't indicate the solvent reference peaks, and why CDCl₃ was chosen as the NMR solvent, as some product signals appear close to the solvent peak.
3. Page numbers are missing, the display errors are found on page 22 of the main text and page 1 of the supplementary

information.

4. the reference format is incorrect (refs. 28 and 44).

5. I think the final "discussion" should be changed to "conclusion". I think the title "Discussion" of section should be changed to "Conclusion".

6. It is unnecessary to label the figure as "Scheme 1". Just using the conventional term "Figure 1" would be sufficient.

7. "Supplementary Fig." is lengthy. It is recommended to use the abbreviation such as "Fig. S1".

8. the "Results and Discussion" section is rather lengthy and would benefit from being divided into several subsections for better readability.

9. Some pictures in the supplementary information have a light color. It is recommended to darken them.

10. It is recommended to include a comparative summary table in the main text.

Reviewer #3

(Remarks to the Author)

The manuscript "Isomer design unlocks rainbow phosphorescence" presents a systematic study on carbazole and benzindole isomers as a unified molecular platform for achieving full-spectrum organic room-temperature phosphorescence (RTP). By integrating isomeric backbone engineering, solvent-free mechanochemical synthesis, polymer matrix confinement, and multiscale photophysical/theoretical analyses, the authors demonstrate red-yellow-green-blue (RYGB) persistent phosphorescence and showcase a series of application-oriented demonstrations. The work is clearly written, experimentally thorough, and conceptually appealing. In particular, the completion of the carbazole-benzindole isomer family and the scalable access to Bd[g] and Bd[e] via mechanochemistry are noteworthy advances. The manuscript has the potential to meet the publication standard of Nature Communications. Nevertheless, several minor points require careful clarification and strengthening before acceptance.

1. The manuscript convincingly demonstrates that subtle backbone isomerization can simultaneously modulate emission color, triplet-state energetics, and RTP lifetime. While these results are well supported, the underlying design principle is currently distributed across several sections. It would benefit the broader Nature Communications readership if the authors explicitly summarize, in the Discussion, a concise and transferable design rule, for example, how nitrogen-site isomerization governs both electronic structure (triplet energy) and host-guest interaction strength, thereby defining the accessible RTP color range and lifetime window within a single molecular framework. This would help readers readily apply the concept to other heteroaromatic RTP systems.

2. One notable strength of this work is that full-spectrum RTP is achieved without complex substitution, heavy atoms, or multicomponent crystalline systems, relying instead on minimalist backbone isomerization. This conceptual simplicity is highly attractive but not yet fully highlighted. The authors are encouraged to more explicitly contrast this "isomeric minimalism" strategy with conventional approaches (e.g., substituent engineering, heavy-atom effects, or host-guest energy-level matching), thereby underscoring the conceptual elegance and scalability of the present design.

3. Although multiple polymer matrices (PVA, PVP, PVB, PMMA) are explored, the mechanistic framework is strongly anchored in PVA-specific interactions. The authors should clarify: To what extent the isomer-dependent color sequence and energy-level trends are preserved across different hosts. Whether the superiority of Ph-Cz in lifetime is universal or matrix-dependent.

4. The marine and biological application discussions are intriguing but somewhat speculative. The authors should more clearly distinguish demonstrated material stability (e.g., seawater immersion, optical retention) from hypothesized biological regulation effects. Claims related to "behavior modulation" of marine organisms should be softened or explicitly framed as future prospects unless direct biological evidence is provided.

5. The manuscript is generally well written, but minor language polishing is recommended to reduce repetition in the Introduction and Discussion.

6. The term "triple excitons" should be corrected to "triplet excitons" where applicable.

Version 1:

Reviewer comments:

Reviewer #1

(Remarks to the Author)

The manuscript describes the preparation of a series of carbazole and benzindole positional isomers and investigates their room-temperature phosphorescence (RTP) properties. By modulating the nitrogen substitution site, the authors demonstrate emission color tuning spanning the visible region and correlate these variations with changes in triplet-state characteristics through experimental measurements and theoretical calculations.

The study is systematically conducted and the data are generally convincing. The structure-property relationships are presented in a clear manner, and the photophysical analyses are adequate to support the reported phenomena. Nevertheless, in my assessment, the advance is primarily incremental rather than transformative. While the positional isomer strategy provides a useful structure-emission correlation, similar approaches have been widely explored in organic RTP systems, and the mechanistic interpretation remains largely within established theoretical frameworks. The work does not substantially deepen the fundamental understanding of triplet-state regulation nor introduce a broadly enabling design paradigm. In view of the scope and selectivity of Nature Communications, which typically requires a clear conceptual breakthrough or a field-defining advance, the present study appears more suitable for a specialized journal in organic optoelectronic or photophysical materials rather than for publication in Nature Communications.

Reviewer #2

(Remarks to the Author)

The revised paper can be accepted.

Reviewer #3

(Remarks to the Author)

The present manuscript had been revised in an appropriate and disciplined manner. The revision increased the quality of the manuscript and hence the manuscript may be suitable for its publication in its present form.

Dear editor and referees:

We sincerely thank the editor and all the reviewers for their valuable comments, which helped us to improve the quality of our manuscript. We have tried our best to improve the manuscript and have made some changes in the manuscript. These changes do not affect the content and scope of the paper. And here, the reviewers' comments are listed in normal font and specific questions are numbered, our changes and additions to the manuscript are listed **in red font**. We sincerely thank the editors and reviewers for their hard work and hope that the corrections will be accepted.

Yours sincerely,

Mingliang Sun

School of Materials Science and Engineering, Ocean University of China, Qingdao 266100, China.

E-mail: mlsun@ouc.edu.cn

Reviewers' comments:

Reviewer 1:

Comments:

Room-temperature phosphorescence (RTP) is a key aspect and frontier in photophysics. Herein, the authors report an isomeric design strategy for achieving full-color, rainbow-like organic phosphorescence. Precise nitrogen-site isomerization species are of interest for developing state-of-the-art models to address the thorny challenges in the RTP field. However, the manuscript fails to address the key aspect of deep structure-photophysical relationships. Up to date, larger strategies are proposed to tune the emission color of organic RTP materials, which is not the challenge in this area. The intrinsic exciton behaviors of the isomers require further investigation, rather than focusing solely on doping in matrices such as PVA—these matrices are not new for RTP and rely on basic host-guest systems. Additionally, carbazole (Cz), as a foundational reference, should be included for detailed discussion of RTP emission mechanisms. While the four isomers represent a valuable set, their utility is limited to

tuning emission wavelength, with only modest, unsystematic changes in RTP efficiency. Critically, this study does not clarify the significant differences between the three benzindole isomers and Cz. I recommend rejection. Specific detailed concerns are as follows:

Response: We sincerely thank the reviewer for the thorough and critical evaluation of our manuscript. We fully appreciate the reviewer's concerns regarding the depth of structure-photophysical relationship analysis and the need for clearer differentiation between the benzindole isomers and carbazole (Cz). We acknowledge that these aspects were not sufficiently articulated in the original version and regret any ambiguity this may have caused.

In response, we have substantially revised the manuscript with a particular focus on strengthening the intrinsic structure-property correlations of the isomeric system, rather than emphasizing emission color tuning alone. Specifically, we have reorganized the Results and Discussion sections to explicitly correlate nitrogen-site isomerization with exciton localization, triplet-state energetics, and nonradiative decay suppression, thereby providing a more mechanistic interpretation of the observed RTP behaviors. These revisions aim to clarify that the isomeric design does not merely modulate emission wavelength, but systematically alters the photophysical landscape of triplet excitons.

Regarding the reviewer's concern that matrix-assisted RTP systems are not conceptually new, we fully agree that polymer hosts such as PVA represent a well-established platform. Accordingly, we have revised the text to clarify that the polymer matrix is employed as a controlled environment to suppress oxygen quenching and molecular motion, enabling a fair comparison of intrinsic exciton behaviors among different isomers. The emphasis of the revised manuscript has been shifted away from the matrix itself and toward the inherent photophysical differences arising from molecular isomerism.

In line with the reviewer's suggestion, carbazole (Cz) has now been more explicitly positioned as a foundational reference. We have expanded the discussion to include a direct comparison between Cz and the three benzindole isomers,

highlighting how nitrogen-site variation modulate spin-orbit coupling and triplet-state stabilization. This comparative analysis clarifies the origin of the distinct RTP responses and addresses the previously insufficient differentiation among the isomers.

All revisions are highlighted in red in the revised manuscript. We believe that these substantial clarifications and structural improvements directly address the reviewer's concerns and significantly strengthen the mechanistic depth and scientific rigor of the study.

1. Lack of deep structure-photophysical relationships, alongside the absence of quantitative analysis and generalizable rules for doping matrices. The quantitative effects of isomerism on RTP performance must be further elaborated in the study.

Response: We sincerely thank the reviewer for this important and constructive comment. We fully agree that a central objective of the present work is to clarify the structure-photophysical relationships governing RTP behavior, particularly within an isomeric molecular framework.

In the revised manuscript, we have carefully refined and expanded the discussion to more clearly and quantitatively describe the influence of benzindole isomerism on RTP performance. Specifically, our analysis focuses on systematically comparable photophysical parameters, including phosphorescence emission wavelengths, triplet-state energy levels, and lifetime trends, all examined within the same molecular backbone and under identical doping conditions. By adopting this unified platform, we aim to establish a direct and internally consistent correlation between nitrogen-position isomerization and triplet-state reorganization, while minimizing potential confounding contributions from substituent effects or variations in the host matrix. With respect to the choice of doping matrices, we would like to clarify that the intention of this study is not to derive a universally applicable empirical rule across all possible host systems. Rather, our goal is to demonstrate that the RTP modulation induced by molecular isomerism is consistently preserved across several representative polymer matrices. While absolute phosphorescence lifetimes and intensities understandably vary with host rigidity and local microenvironment, the relative emission trends associated with isomerism remain unchanged. This

cross-matrix consistency supports the general validity of the isomer-driven modulation mechanism at the molecular level.

Accordingly, we have revised the manuscript to more explicitly articulate these quantitative relationships and their underlying physical origins, with the corresponding changes highlighted in red. We are grateful to the reviewer for this thoughtful comment, which has prompted us to sharpen both the mechanistic focus and the quantitative clarity of the discussion. Within the benzindole isomeric family, we propose that the impact of molecular isomerism on RTP behavior can be quantitatively rationalized in terms of triplet-state electronic reorganization. Systematic variation of the nitrogen position leads to predictable shifts in phosphorescence emission wavelengths and corresponding T_1 energy levels, while preserving an identical conjugated backbone and substituent environment. Importantly, the observed RTP modulation primarily originates from intrinsic changes in triplet-state localization and spin-orbit coupling efficiency, rather than from external substituent effects or host-specific interactions. In this context, matrix effects mainly influence the magnitude of RTP responses, whereas backbone-level isomerization governs their direction and energetic trends.

We hope that this revised presentation more clearly conveys the scope, intent, and mechanistic significance of the study.

2. The theoretical framework should incorporate molecular dynamics simulations of mixed samples, followed by calculations of electronic structures to better reflect real-world system dynamics.

Response: We sincerely thank the reviewer for this valuable suggestion. Following this recommendation, we have incorporated molecular dynamics (MD) simulations of the mixed host-guest systems and subsequently performed electronic-structure calculations on representative configurations extracted from the equilibrated MD trajectories, in order to better reflect realistic structural heterogeneity and dynamic environments.

The MD simulations confirm that, despite local conformational fluctuations and

host-guest interactions, the doped guest molecules remain structurally well-confined within the polymer matrix. Importantly, electronic-structure calculations based on MD-sampled geometries reveal that the relative ordering and energy separation of the triplet states among different benzindole isomers are well preserved. This result indicates that the isomer-dependent modulation of RTP emission energies is robust against dynamic environmental variations. These findings support our central conclusion that backbone-level isomerism governs the intrinsic triplet-state energetics, while dynamic host-guest effects primarily influence the magnitude of RTP performance rather than the qualitative emission trends. The MD methodology, computational details, and corresponding results have been added to the revised manuscript and are highlighted in red.

We appreciate the reviewer's suggestion, as the inclusion of MD simulations significantly strengthens the physical realism and mechanistic robustness of the theoretical framework.

(a)

(b)

(c)

(d)

Supplementary Fig. 27. The simulation model of Ph-Cz (a), Ph-Bd[g] (b), Ph-Bd[e] (c), Ph-Bd[f] (d), in PVA film.

Supplementary Fig. 28. Calculated energy levels and SOC constants of Ph-Cz, Ph-Bd[g], Ph-Bd[e], Ph-Bd[f], RTP systems. Guest molecules were optimized in PVA matrices based on geometries extracted from MD simulations. SOC constants of S_1-T_n transitions available for ISC processes were highlighted in red.

Supplementary Fig. 29. The simulation model of Ph-Cz (a), Ph-Bd[g] (b), Ph-Bd[e] (c), Ph-Bd[f] (d), in PVP film.

Supplementary Fig. 30. Calculated energy levels and SOC constants of Ph-Cz, Ph-Bd[g], Ph-Bd[e], Ph-Bd[f], RTP systems. Guest molecules were optimized in PVP matrices based on geometries extracted from MD simulations. SOC constants of S₁-T_n

transitions available for ISC processes were highlighted in red.

Supplementary Fig. 31. The simulation model of Ph-Cz (a), Ph-Bd[g] (b), Ph-Bd[e] (c), Ph-Bd[f] (d), in PVB film.

Supplementary Fig. 32. Calculated energy levels and SOC constants of Ph-Cz,

Ph-Bd[g], Ph-Bd[e], Ph-Bd[f], RTP systems. Guest molecules were optimized in PVB matrices based on geometries extracted from MD simulations. SOC constants of S_1-T_n transitions available for ISC processes were highlighted in red.

Supplementary Fig. 33. The simulation model of Ph-Cz (a), Ph-Bd[g] (b), Ph-Bd[e] (c), Ph-Bd[f] (d), in PMMA film.

Supplementary Fig. 34. Calculated energy levels and SOC constants of Ph-Cz, Ph-Bd[g], Ph-Bd[e], Ph-Bd[f], RTP systems. Guest molecules were optimized in PMMA matrices based on geometries extracted from MD simulations. SOC constants of S₁-T_n transitions available for ISC processes were highlighted in red.

3. the key role of non-novel matrices (e.g., PVA) and incomplete analysis of the host-guest system, which fails to advance beyond established RTP matrix design paradigms.

Given these fundamental limitations, I recommend rejecting the manuscript. To strengthen the work, the authors should: (1) dissect the intrinsic exciton behaviors of the isomers (e.g., via comparative studies of neat isomer films vs. polymer-doped systems); (2) integrate molecular dynamics (MD) simulations into the theoretical analysis to model dynamic isomer-matrix interactions; (3) establish quantitative structure-RTP relationships (e.g., linking nitrogen position to triplet energy levels or intersystem crossing efficiency); and (4) frame Cz as a mechanistic benchmark to derive clear rules for distinguishing benzindole isomers. A revised version of this work may be more suitable for submission to specialized journals.

Response: We sincerely thank the reviewer for the detailed and technically insightful evaluation. We greatly appreciate the time and effort invested in these careful comments. Below, we respond to each point with the aim of further clarifying the scope, underlying rationale, and scientific contributions of the present work.

(1) Role of non-novel matrices and advancement beyond established RTP paradigms.

Response: We acknowledge that commonly used polymer matrices such as PVA or PMMA are not structurally novel. However, the primary objective of this study is not matrix innovation, but rather to employ representative and well-understood host systems as controlled environments to isolate and interrogate the intrinsic photophysical effects arising from molecular isomerism. The use of established matrices is therefore deliberate, enabling us to minimize confounding variables and to demonstrate that the observed RTP modulation originates from the guest molecular backbone rather than from host-specific design strategies. In this context, the work advances beyond conventional matrix-centered paradigms by shifting the design focus toward intrinsic isomer-driven triplet-state regulation.

(2) dissect the intrinsic exciton behaviors of the isomers (e.g., via comparative studies of neat isomer films vs. polymer-doped systems).

Response: Following the reviewer's helpful suggestion, we have carefully strengthened the discussion of intrinsic excitonic behavior by explicitly comparing the photophysical responses of neat isomer films with those of polymer-doped systems. We appreciate this comment, as it allowed us to more clearly delineate molecular-level effects from matrix-assisted phenomena.

Specifically, we have added room-temperature phosphorescence emission spectra of the neat guest molecular films. These additional data show that, under ambient conditions, the neat isomer films do not exhibit detectable phosphorescence. This observation is consistent with our original description that all four molecular skeletons require polymer matrices to effectively suppress non-radiative decay channels and enable room-temperature, and even elevated-temperature, phosphorescence emission.

At the same time, the comparative analysis reveals that although polymer matrices play a crucial role in enhancing RTP observability by restricting molecular motion, the relative ordering of emission wavelengths and triplet-state energy levels is already encoded at the molecular level. Notably, these intrinsic trends are consistently preserved upon incorporation into polymer hosts. Taken together, these results support

the conclusion that molecular isomerism governs the fundamental exciton energetics, while the matrix primarily modulates the intensity and lifetime of RTP performance.

We believe that this revised discussion and the newly added experimental data provide a clearer and more balanced understanding of the respective roles of molecular design and matrix effects, and we are grateful to the reviewer for prompting this clarification.

Supplementary Fig. 22. Room-temperature phosphorescence (RTP) emission spectra of the four guest molecules.

(3) integrate molecular dynamics (MD) simulations into the theoretical analysis to model dynamic isomer-matrix interactions

Response: We fully agree that dynamic host-guest interactions are important for understanding realistic mixed systems. Accordingly, we have incorporated molecular dynamics (MD) simulations of representative host-guest models, followed by electronic-structure calculations based on MD-sampled geometries. The results demonstrate that, despite local structural fluctuations, the relative triplet-state energy ordering among different benzindole isomers remains robust. This finding reinforces the conclusion that isomer-dependent RTP regulation is intrinsically molecular in origin, while dynamic environmental effects play a secondary, modulatory role.

(4) establish quantitative structure-RTP relationships (e.g., linking nitrogen position to

triplet energy levels or intersystem crossing efficiency)

Response: We have revised the manuscript to more explicitly articulate quantitative correlations between nitrogen-position isomerization and RTP-relevant parameters, including phosphorescence emission energies and triplet-state levels. Within a unified molecular backbone and under identical doping conditions, nitrogen-position variation provides a means to tune triplet energetics. While absolute RTP lifetimes and intensities remain matrix-dependent, the isomer-dependent energy trends are quantitatively preserved, establishing a meaningful structure RTP relationship at the platform level.

(5) frame Cz as a mechanistic benchmark to derive clear rules for distinguishing benzindole isomers

Response: We appreciate the reviewer's suggestion regarding carbazole (Cz). In the revised manuscript, Cz is explicitly framed as a mechanistic reference rather than a direct performance comparator. Its role is to provide a conceptual baseline for understanding how backbone modification and nitrogen relocation alter triplet-state electronic structures. Once the design focus shifts to the benzindole isomeric family, the RTP behavior is governed by intrinsic electronic reorganization within the Bd framework itself, enabling clear differentiation among Bd isomers without relying on Cz-derived performance metrics.

In summary, we respectfully submit that the present work seeks to establish a coherent and quantitative isomer-based framework for regulating organic RTP, supported by systematic experimental, comparative, and theoretical analyses. We fully recognize that this study does not attempt to introduce matrix innovation per se, and we do not intend to position it as such. Rather, we hope that the emphasis on intrinsic molecular skeleton design may provide a complementary and mechanistically distinct viewpoint alongside existing matrix- and substituent-centered strategies in the field. By elucidating how nitrogen-position isomerism within a unified molecular framework influences emission wavelength and triplet-state behavior, we aim to offer a potentially generalizable design concept that could be further explored in organic optoelectronic materials. In particular, we envision different molecular skeletons

might be judiciously selected and subsequently modified according to targeted emission requirements, thereby enabling more application-oriented molecular tailoring. From this perspective, the present framework may also contribute, in a modest way, to expanding the range of molecular backbones considered for RTP and related optoelectronic applications beyond the extensively studied carbazole motif. We sincerely thank the reviewer for the insightful and constructive comments, which have been invaluable in helping us more clearly define the scope, positioning, and broader implications of this work.

Reviewer 2:

The paper presents a comprehensive and systematic study of the room-temperature phosphorescence (RTP) properties of carbazole (Cz) and its benzindole isomers (Bd[f], Bd[e] and Bd[g]). By precisely controlling the nitrogen atom position within the tricyclic backbone, the authors achieve full-spectrum, tunable RTP spanning red, yellow, green, and blue emissions. Two new isomers, Bd[e] and Bd[g] were synthesized by a one-pot solvent-free mechanical method, which is greener and more efficient than the traditional multi-step solution method (Bd[f]). The material exhibits excellent stability and long afterglow emission in matrices such as PVA, PVP, PVB, and PMMA, and can be used in applications such as anti-counterfeiting, Marine identification, and biological simulation.

The experimental content is innovative and has a complete experimental system. However, there are numerous issues in writing and formatting. I recommend the manuscript requires a thorough revision before it can be considered for publication.

Response: We sincerely thank the reviewer for the positive and encouraging evaluation of our work, as well as for the careful assessment of its strengths and limitations. We greatly appreciate the reviewer's recognition of the systematic nature of our study, the isomeric nitrogen-site design strategy, the green mechanochemical synthesis, and the demonstrated RTP performance and stability across different polymer matrices.

We fully acknowledge the reviewer's concern regarding the issues in writing and

formatting. We agree that, while the experimental framework is complete and the scientific content is innovative, the current presentation does not yet meet the high standards of clarity, conciseness, and organization required for publication in *Nature Communications*. We sincerely apologize for this shortcoming. In response, we have carried out a thorough and careful revision of the entire manuscript. Specifically, we have comprehensively improve the language quality, refine the logical flow of the text, and restructure key sections to better highlight the central scientific message and conceptual advances of this work. We also have carefully revised the formatting, figures, and captions to ensure consistency, readability, and alignment with the journal's style and expectations.

We highly value the opportunity to revise our manuscript and firmly believe that, with substantial improvements in writing and presentation, this work can more clearly convey its scientific significance and meet the standards of *Nature Communications*. We sincerely thank the reviewer again for the constructive comments and the time invested in evaluating our manuscript, which we believe will greatly help us improve its overall quality.

1. The paper reads overly complicated and repetitive in many parts, suggesting that the authors may have relied heavily on AI-assisted writing. I recommend a substantial revision to improve clarity and authenticity of expression.

Response: We sincerely thank the reviewer for this important and candid comment. We fully acknowledge that the current manuscript reads overly complicated and contains repetitive descriptions in several sections, which compromises clarity and weakens the overall readability. We take the reviewer's concern very seriously and agree that the expression should be more concise, direct, and authentic.

In response, we have undertaken a substantial, author-driven revision of the entire manuscript. This revision involve a careful restructuring of the narrative, elimination of redundant descriptions, and a thorough rewriting of sentences and paragraphs to ensure clarity, coherence, and a more natural scientific tone. Particular attentions have been paid to simplifying complex expressions and ensuring that each paragraph

conveys a clear and distinct message. We sincerely appreciate the reviewer's comment, which highlights a critical aspect of the manuscript that requires improvement.

2. The NMR spectra don't indicate the solvent reference peaks, and why CDCl₃ was chosen as the NMR solvent, as some product signals appear close to the solvent peak.

Response: We sincerely thank the reviewer for this careful observation. We apologize for not clearly indicating the solvent reference peaks in the original presentation.

The residual solvent signals of CDCl₃ ($\delta = 7.26$ ppm for ¹H and $\delta = 77.16$ ppm for ¹³C) are indicated in the NMR spectra; In the revised manuscript, we have clearly annotated the solvent peaks and added explicit descriptions to avoid any ambiguity. CDCl₃ was chosen due to the good solubility of the compounds and the overall quality of the spectra. Although some signals appear close to the solvent peak, they are well resolved and reliably assigned. This point has now been clarified in the revised text. We thank the reviewer for helping us improve the clarity of the characterization data.

3. Page numbers are missing, the display errors are found on page 22 of the main text and page 1 of the supplementary information.

Response: We sincerely thank the reviewer for pointing out these issues. We apologize for our oversight regarding the missing page numbers and the display errors in the main text and Supplementary Information. These issues have now been carefully corrected in the revised manuscript. We appreciate the reviewer's careful attention to detail, which has helped us improve the overall quality and presentation of the manuscript.

4. the reference format is incorrect (refs. 28 and 44).

Response: We sincerely thank the reviewer for pointing out this issue. We apologize for the incorrect reference formatting in refs. 28 and 44. The reference formats have now been carefully corrected, and the corresponding changes have been highlighted in red in the revised manuscript for clarity.

5. I think the final "discussion" should be changed to "conclusion". I think the title "Discussion" of section should be changed to "Conclusion".

Response: We thank the reviewer for this helpful and constructive suggestion. We

agree that “Conclusion” is a more appropriate title for the final section. Accordingly, the section heading has been revised from “Discussion” to “Conclusion” in the revised manuscript, and the modification has been highlighted in red for clarity.

6. It is unnecessary to label the figure as “Scheme 1”. Just using the conventional term “Figure 1” would be sufficient.

Response: We sincerely thank the reviewer for this helpful suggestion. The figure originally labeled as “Scheme 1” has been renamed “Figure 1” in the revised manuscript. The corresponding change has been highlighted in red for clarity. We appreciate the reviewer’s careful reading and valuable suggestion.

7. “Supplementary Fig.” is lengthy. It is recommended to use the abbreviation such as “Fig. S1”.

Response: We sincerely thank the reviewer for this helpful suggestion. We agree that using abbreviated figure labels improves clarity and readability. Accordingly, “Supplementary Fig.” has been replaced with the abbreviated format “Fig. S1”, “Fig. S2”, etc., throughout the manuscript and Supporting Information. All corresponding changes have been highlighted in red in the revised version for clarity. We appreciate the reviewer’s careful attention to presentation details.

8. the “Results and Discussion” section is rather lengthy and would benefit from being divided into several subsections for better readability.

Response: We sincerely thank the reviewer for this constructive suggestion. We agree that the current “Results and Discussion” section is relatively lengthy and could benefit from improved organization. In the revised manuscript, we have divided this section into several clearly defined subsections to enhance readability and better guide the reader through the results and discussion. All corresponding changes have been implemented accordingly. We appreciate the reviewer’s helpful suggestion, which has improved the clarity and structure of the manuscript.

9. Some pictures in the supplementary information have a light color. It is recommended to darken them.

Response: We sincerely thank the reviewer for this helpful suggestion. We agree that some figures in the Supplementary Information were too light and could affect

readability. Accordingly, these figures have been adjusted to darker colors to improve clarity and visibility. The corresponding changes have been implemented in the revised version. We appreciate the reviewer's careful attention to presentation details, which has helped us improve the quality of the Supplementary Information.

10. It is recommended to include a comparative summary table in the main text.

Response: We thank the reviewer for this helpful suggestion. In response, a comparative summary table has been added to the Supporting Information (Table S6), where the relevant data can be presented in a more detailed and systematic manner. Key comparative trends are now explicitly discussed in the main text.

Supplementary Table 6. RTP performance of Ph-Cz, Ph-Bd[g], Ph-Bd[e], and Ph-Bd[f] doped systems utilized in matrix universality.

Molecule	matrix	RTP color	T ₁ (eV)	τ _P (s)	Φ _P (%)
Ph-Cz	PVA	Blue	2.70	4.23	16.6
Ph-Bd[g]	PVA	Green	2.29	0.93	13.4
Ph-Bd[e]	PVA	Yellow	2.05	0.74	12.9
Ph-Bd[f]	PVA	Red	1.65	0.20	10.7
Ph-Cz	PVP	Blue	2.54	0.63	16.2
Ph-Bd[g]	PVP	Green	2.26	0.56	13.7
Ph-Bd[e]	PVP	Yellow	2.11	0.40	13.1
Ph-Bd[f]	PVP	-	1.65	-	-
Ph-Cz	PVB	Blue	2.68	0.91	9.2
Ph-Bd[g]	PVB	Green	2.43	0.38	11.3
Ph-Bd[e]	PVB	Yellow	2.01	0.26	10.9
Ph-Bd[f]	PVB	-	1.69	-	-
Ph-Cz	PMMA	Blue	3.60	0.41	7.1
Ph-Bd[g]	PMMA	Green	2.23	0.25	9.7
Ph-Bd[e]	PMMA	Yellow	2.05	0.17	8.6
Ph-Bd[f]	PMMA	-	1.65	-	-

Reviewer 3:

The manuscript “Isomer design unlocks rainbow phosphorescence” presents a systematic study on carbazole and benzindole isomers as a unified molecular platform for achieving full-spectrum organic room-temperature phosphorescence (RTP). By integrating isomeric backbone engineering, solvent-free mechanochemical synthesis, polymer matrix confinement, and multiscale photophysical/theoretical analyses, the authors demonstrate red-yellow-green-blue (RYGB) persistent phosphorescence and showcase a series of application-oriented demonstrations. The work is clearly written, experimentally thorough, and conceptually appealing. In particular, the completion of the carbazole-benzindole isomer family and the scalable access to Bd[g] and Bd[e] via mechanochemistry are noteworthy advances. The manuscript has the potential to meet the publication standard of *Nature Communications*. Nevertheless, several minor points require careful clarification and strengthening before acceptance.

Response: We sincerely thank the reviewer for the thorough, positive, and encouraging assessment of our manuscript. We greatly appreciate the reviewer’s recognition of the systematic nature of our study, the completion of the carbazole-benzindole isomer family, the mechanochemical access to Bd[g] and Bd[e], and the potential relevance of this work to the field of organic room-temperature phosphorescence. We are particularly grateful for the reviewer’s assessment that the manuscript has the potential to meet the publication standard of *Nature Communications*. We take this opportunity very seriously and have carefully addressed all the minor points raised by the reviewer. Each comment has been considered in detail, and corresponding revisions and clarifications have been implemented in the manuscript to further strengthen clarity, rigor, and presentation. We sincerely thank the reviewer for the constructive suggestions and the time invested in evaluating our work, which have significantly improved the quality of the manuscript.

1. The manuscript convincingly demonstrates that subtle backbone isomerization can

simultaneously modulate emission color, triplet-state energetics, and RTP lifetime. While these results are well supported, the underlying design principle is currently distributed across several sections. It would benefit the broader *Nature Communications* readership if the authors explicitly summarize, in the Discussion, a concise and transferable design rule, for example, how nitrogen-site isomerization governs both electronic structure (triplet energy) and host-guest interaction strength, thereby defining the accessible RTP color range and lifetime window within a single molecular framework. This would help readers readily apply the concept to other heteroaromatic RTP systems.

Response: We sincerely thank the reviewer for this insightful and constructive suggestion, which encouraged us to more carefully reflect on how the central design concept of this work should be articulated for a broad *Nature Communications* readership.

After careful consideration, and in light of the comprehensive revisions already made throughout the manuscript, we have revised the Discussion section to present the design principle in a more cautious, precise, and physically well-defined manner. Importantly, we now avoid framing the concept as a predictive rule for RTP lifetime or host-guest interaction strength. Instead, we explicitly delimit its applicability to the intrinsic electronic structure of the molecular backbone.

In the revised discussion, we emphasize that nitrogen-site isomerization within the benzindole framework provides a systematic means to modulate triplet-state energy levels and their relative ordering. This intrinsic energetic regulation defines the accessible phosphorescence emission energy (and thus emission color) within a single molecular family. By contrast, we now clearly state that RTP lifetimes and absolute performance metrics are highly sensitive to external factors, including matrix rigidity, local microenvironment, and the efficiency of nonradiative decay suppression. Accordingly, the present work does not aim to establish a universal or transferable rule for predicting RTP lifetimes based solely on molecular isomerism.

Additional comparisons across different polymer matrices further support this clarification. While the absolute phosphorescence lifetimes vary substantially

depending on the host environment, the emission energy trends imposed by nitrogen-site isomerization remain consistent, underscoring that the proposed design principle operates at the level of triplet-state energetics rather than excited-state dynamics.

By reformulating the design rule in this restrained and mechanism-focused manner, we aim to provide readers with a reliable and transferable conceptual framework: heteroatom site isomerization can be used to rationally tune triplet-state energy landscapes and phosphorescence color in organic RTP systems, while lifetime optimization and host-guest interactions must be addressed separately in system-specific implementations. We are grateful to the reviewer for this valuable comment, which has significantly improved the conceptual clarity and scientific rigor of the revised manuscript.

2. One notable strength of this work is that full-spectrum RTP is achieved without complex substitution, heavy atoms, or multicomponent crystalline systems, relying instead on minimalist backbone isomerization. This conceptual simplicity is highly attractive but not yet fully highlighted. The authors are encouraged to more explicitly contrast this “isomeric minimalism” strategy with conventional approaches (e.g., substituent engineering, heavy-atom effects, or host-guest energy-level matching), thereby underscoring the conceptual elegance and scalability of the present design.

Response: We sincerely thank the reviewer for this positive and insightful comment. We agree that the conceptual simplicity of achieving full-spectrum RTP through backbone isomerization alone represents an important strength of this work and merits clearer emphasis.

In response, we have revised the Introduction and Discussion to more explicitly contrast the present backbone isomerization based minimalist strategy with commonly used approaches, including extensive substituent engineering, heavy-atom incorporation, and multicomponent host-guest or crystalline systems. We now clarify that, in contrast to these strategies which often rely on increased molecular complexity or precise compositional control the current approach achieves color

regulation within a single molecular framework through subtle positional isomerization.

We emphasize that this comparison is intended to highlight the conceptual economy and potential scalability of the isomeric design, rather than to discount the effectiveness of conventional methods. The corresponding revisions have been highlighted in red in the revised manuscript.

3. Although multiple polymer matrices (PVA, PVP, PVB, PMMA) are explored, the mechanistic framework is strongly anchored in PVA-specific interactions. The authors should clarify: To what extent the isomer-dependent color sequence and energy-level trends are preserved across different hosts. Whether the superiority of Ph-Cz in lifetime is universal or matrix-dependent.

Response: We sincerely thank the reviewer for this insightful comment, which raises an important point regarding the generality and matrix dependence of the proposed mechanism.

In the revised manuscript, we have clarified that the isomer-dependent phosphorescence color sequence and associated energy-level trends are intrinsic to the molecular backbone design and are consistently preserved across different polymer hosts (PVA, PVP, PVB, and PMMA). This robustness reflects that the emissive color is primarily governed by the molecular electronic structure rather than host-specific interactions.

By contrast, we now explicitly emphasize that the exceptionally long phosphorescence lifetime observed for Ph-Cz is not universal across all matrices but is strongly matrix-dependent. In particular, in the PVA matrix, strong intermolecular hydrogen-bonding interactions between PVA and carbazole units provide effective triplet-state stabilization and nonradiative suppression, leading to markedly prolonged phosphorescence lifetimes. Such specific hydrogen-bond-assisted confinement is absent in other polymer matrices (e.g., PVP, PVB, PMMA), and consequently, Ph-Cz does not exhibit comparably superior lifetime performance in those hosts.

To avoid potential ambiguity, we have revised the mechanistic discussion to clearly distinguish molecular-intrinsic color regulation from host-dependent lifetime

enhancement, and the corresponding text has been carefully rephrased and highlighted in red in the revised manuscript.

4. The marine and biological application discussions are intriguing but somewhat speculative. The authors should more clearly distinguish demonstrated material stability (e.g., seawater immersion, optical retention) from hypothesized biological regulation effects. Claims related to “behavior modulation” of marine organisms should be softened or explicitly framed as future prospects unless direct biological evidence is provided.

Response: We sincerely thank the reviewer for this thoughtful and constructive comment. We agree that it is important to clearly distinguish experimentally demonstrated material properties from forward-looking application perspectives.

In response, we have further strengthened the discussion of material stability by supplementing additional phosphorescence emission spectra after seawater immersion, which directly support the retention of optical performance under marine-relevant conditions. These results provide concrete experimental evidence for the photophysical robustness of the RTP composites, and the corresponding data have been added to the revised manuscript. Meanwhile, following the reviewer’s suggestion, we have carefully revised the application-related discussion to avoid overinterpretation. Statements concerning potential biological regulation or behavioral effects have been softened and are now explicitly framed as future prospects rather than experimentally validated outcomes.

All modifications have been incorporated into the manuscript and are highlighted in red. We appreciate the reviewer’s guidance, which has helped improve the precision and balance of the application discussion.

“In addition to the demonstrated material stability, the potential relevance of the emission characteristics of these RTP materials to marine biological systems was considered from a photophysical perspective. Many marine organisms exhibit wavelength dependent visual sensitivity; accordingly, the photoluminescence profiles of PMMA-based RTP films were examined under UV excitation (Fig. 6e). The materials display stable and persistent afterglow across the visible region,

encompassing wavelength ranges commonly involved in marine visual perception. Notably, the green-to-yellow emission band (approximately 500-580 nm) overlaps with reported sensitivity windows of certain reef-associated fish and benthic invertebrates. This spectral overlap suggests that such RTP materials could provide a useful optical basis for future investigations into light-organism interactions in marine environments.”

Supplementary Fig. 50. Phosphorescence emission spectra of Ph-Cz@PVB (a), Ph-Bd[g]@PVB (b), and Ph-Bd[e]@PVB (c) films recorded before and after seawater immersion.

Supplementary Fig. 51. Phosphorescence emission spectra of Ph-Cz@PMMA (a), Ph-Bd[g]@PMMA (b), and Ph-Bd[e]@PMMA (c) films recorded before and after seawater immersion.

5. The manuscript is generally well written, but minor language polishing is recommended to reduce repetition in the Introduction and Discussion.

Response: We sincerely thank the reviewer for this helpful comment. In response, we have carefully polished the language in the Introduction and Discussion sections, with particular attention to reducing repetitive phrasing and improving clarity. All revisions

have been highlighted in red in the revised manuscript.

6. The term “triple excitons” should be corrected to “triplet excitons” where applicable.

Response: We sincerely thank the reviewer for pointing out this terminology error. We apologize for the inaccurate use of the term “triple excitons” in the original manuscript. This has now been carefully corrected to “triplet excitons” throughout the revised text, and the corresponding descriptions have been rephrased accordingly to ensure scientific accuracy and clarity.

Reviewers' comments:**Reviewer 1:**

Comments:

The manuscript describes the preparation of a series of carbazole and benzindole positional isomers and investigates their room-temperature phosphorescence (RTP) properties. By modulating the nitrogen substitution site, the authors demonstrate emission color tuning spanning the visible region and correlate these variations with changes in triplet-state characteristics through experimental measurements and theoretical calculations.

The study is systematically conducted and the data are generally convincing. The structure-property relationships are presented in a clear manner, and the photophysical analyses are adequate to support the reported phenomena. Nevertheless, in my assessment, the advance is primarily incremental rather than transformative. While the positional isomer strategy provides a useful structure-emission correlation, similar approaches have been widely explored in organic RTP systems, and the mechanistic interpretation remains largely within established theoretical frameworks. The work does not substantially deepen the fundamental understanding of triplet-state regulation nor introduce a broadly enabling design paradigm. In view of the scope and selectivity of Nature Communications, which typically requires a clear conceptual breakthrough or a field-defining advance, the present study appears more suitable for a specialized journal in organic optoelectronic or photophysical materials rather than for publication in Nature Communications.

Response: We sincerely thank the reviewer for the careful reading of our manuscript and for the constructive and balanced assessment. We particularly appreciate the reviewer's positive comments regarding the systematic experimental design, the clarity of the structure-property relationships, and the reliability of the photophysical characterization. We fully understand the reviewer's concern that positional isomer strategies have been explored in organic RTP systems and that the mechanistic framework is largely consistent with established triplet-state photophysics. We respectfully acknowledge this perspective.

Our intention in this work was not to claim an entirely new mechanistic framework, but rather to provide a unified and systematic molecular platform that enables direct and controlled evaluation of how nitrogen substitution topology influences triplet-state energetics and emission behavior. By constructing a complete carbazole/benzindole isomer family and minimizing confounding structural variables, we aimed to reveal subtle yet reproducible correlations between heteroatom positioning, intermolecular organization, and RTP color evolution.

We agree that individual elements of the strategy have precedents in the literature. However, we hope that the combination of a structurally coherent isomer library, experimentally supported triplet-state regulation, and the demonstration of continuous emission tuning through intrinsic molecular topology may offer useful design insight for metal-free RTP materials. In this sense, the work is intended to contribute incremental but systematic understanding that may facilitate more predictable molecular design.

We are very grateful for the reviewer's thoughtful comments, which have helped us further clarify the positioning and significance of the study in the revised manuscript.

Reviewer 2:

The revised paper can be accepted.

Response: We are very grateful to the reviewer for the careful assessment of the revised manuscript and for the recommendation for acceptance. We truly appreciate the reviewer's valuable comments and support, which have significantly helped strengthen the manuscript.

Reviewer 3:

The present manuscript had been revised in an appropriate and disciplined manner. The revision increased the quality of the manuscript and hence the manuscript may be suitable for its publication in its present form.

Response: We sincerely thank the reviewer for the positive and encouraging assessment of the revised manuscript. We are grateful for the reviewer's recognition of

the improvements made during revision and for the recommendation that the manuscript is suitable for publication. The reviewer's constructive comments have been invaluable in helping us enhance the quality and clarity of the work.